# Mycobiomes of two distinct clades of ambrosia gall midges (Diptera: Cecidomyiidae) are species-specific in larvae but similar in nutritive mycelia

Petr Pyszko,[1] Hana Šigutová,[1,2] Miroslav Kolařík,[3] Martin Kostovčík,[3] Jan Ševčík,[1] Martin Šigut,[1,3] Denisa Višňovská,[1,3] Pavel Drozd[1]

**ABSTRACT**    Ambrosia gall midges (AGMs) are mostly host plant-specific. In their galls, they harbor fungal symbionts on which they feed. Therefore, they represent unique steps in the evolution of the gall-forming Cecidomyiidae (Diptera). Gall-associated fungi have been studied predominantly by cultivations, and potential larval endosymbionts have been completely neglected. Using ITS2 rRNA metabarcoding, we characterized the mycobiomes of individual gall compartments (gall surface, gall interior, and larva) of six species from two phylogenetically separated tribes (Asphondyliini and Lasiopterini). Compared to the gall surface and interior, the larvae harbored significantly higher fungal richness and taxonomic diversity, and a larger pool of indicator taxa. Larval mycobiome composition was more species-specific; however, the fungal genera *Fusarium*, *Filobasidium*, *Tilletiopsis*, *Alternaria*, and *Aureobasidium* were indicator taxa shared among species. Overall, the larvae harbored 29% of unique taxa that can play a functional role in the host (e.g., initiation of gall development or selection of the mycelia composition). The mycobiome of the gall interior was assembled least stochastically, and its composition was the least species-specific, being dominated by *Botryosphaeria dothidea* (except for *Lasioptera arundinis*). Therefore, the interior of ambrosia galls offers a unique environment that supports the growth of similar fungi, regardless of the host plant species and the phylogenetic distance between the AGM tribes. Our study illustrates a range of fungal microorganisms indicative of individual gall compartments, but their potential function, especially in larvae, remains to be solved.

**IMPORTANCE**    Ambrosia gall midges are endophagous insect herbivores whose larvae live enclosed within a single gall for their entire development period. They may exhibit phytomycetophagy, a remarkable feeding mode that involves the consumption of plant biomass and mycelia of their cultivated gall symbionts. Thus, AGMs are ideal model organisms for studying the role of microorganisms in the evolution of host specificity in insects. However, compared to other fungus-farming insects, insect–fungus mutualism in AGMs has been neglected. Our study is the first to use DNA metabarcoding to characterize the complete mycobiome of the entire system of the gall-forming insects as we profiled gall surfaces, nutritive mycelia, and larvae. Interestingly, larval mycobiomes were significantly different from their nutritive mycelia, although *Botryosphaeria dothidea* dominated the nutritive mycelia, regardless of the evolutionary separation of the tribes studied. Therefore, we confirmed a long-time hypothesized paradigm for the important evolutionary association of this fungus with AGMs.

**KEYWORDS**    ambrosia gall midge, Cecidomyiidae, fungiculture, *Asphondylia*, *Lasioptera*, larval mycobiome, metabarcoding, nutritive mycelium, phytomycetophagy

Address correspondence to Petr Pyszko, petr.pyszko@osu.cz.

The authors declare no conflict of interest.

See the funding table on p. 12.

10.1128/spectrum.02830-23 **1**

Ambrosia gall midges (AGMs) are mostly host plant-specific and feed on the mycelia of the cultivated fungal symbiont(s). Due to their mixed diet (phytomycetophagy), this group represents a unique step in the evolution of gall-forming Cecidomyiidae (1). Most AGMs belong to two taxonomically separated tribes of the subfamily Cecidomyiinae: Asphondyliini, and Lasiopterini (2–5). Fungiculture, which has evolved independently at least six times in insects (6), has been intensively studied in bark and ambrosia beetles (7), woodwasps (8, 9), leaf-cutting ants (10), and fungus-growing termites (11); however, in AGMs, this interaction has received far less attention.

Based on current knowledge, Asphondyliini may be associated with specific fungal symbionts, primarily *Botryosphaeria dothidea* (2, 12–15). However, this fungus has also been found in other AGM-induced galls, including Lasiopterini (13, 16–18), and also in the unrelated Cynipidae (Hymenoptera) (19, sequence accession KT823763), where fungi as a diet are not expected. Moreover, a variety of other fungi have been found in the galls of *Asphondylia* (2, 12, 20, 21), *Lasioptera* (22, 23), and other AGM genera (13). However, due to the lack of experimental design in most studies, these fungi have been reported to be "associated" rather than "mutualists" and sometimes even as incidentally captured saprotrophs (17, 24). Therefore, the specificity and richness of fungal symbionts in ambrosia galls remain largely speculative.

Although insect fungicultures are usually complex (6), fungal associates of AGMs have been studied predominantly by cultivation, sometimes complemented by standard DNA barcoding (12, 13, 17, 18, 20, 22, 25–28). In addition to capturing only a fragment of diversity, gall-based cultivations are prone to contamination (29) and fail frequently, especially for dominant *B. dothidea* (13, 16). Some fungi present in ambrosia galls may be only saprophytes, representing opportunistic colonization rather than strict mutualism (12). These species frequently reported as gall associates may conceal the real symbiont(s) during the isolation procedure (27). These shortcomings can be overcome by the application of culture-independent methods and the inclusion of control samples, for example, tissues of gall surfaces or ungalled plant parts.

In AGMs, mycobiome studies have focused primarily on nutritive mycelia (14, 18). On occasion, fungal samples were collected from the external surfaces of the galls (30), ungalled plant parts (20), eggs (17), and larval surfaces (13). However, a comparison of the respective gall compartments (i.e., gall surface, interior, and larvae) has never been made. Despite the urgency to detect fungal DNA from larvae (13), this has never been done, and it remains an open question whether there is a distinct mycobiome of the individuals that live permanently inside the gall enclosed by their nutritive mycelia. In herbivorous or detritivorous insects, fungi represent an important part of the microbiome, often forming communities distinct from those present in the feeding substrates (31–35). Paradoxically, in fungus-feeding insects, only the bacterial component of the gut has been studied (36–40), except for ambrosia beetles (41, 42). Therefore, the potential role of fungi in AGM larvae remains unknown.

In this study, we sampled galls from six AGM species from two phylogenetically separated Cecidomyiidae tribes (Asphondyliini and Lasiopterini). We used DNA metabarcoding to profile the fungal communities of each compartment of the whole AGM system: external gall surface (plant tissue), nutritive mycelia (gall interior), and larvae. We aimed to (i) investigate the effect of AGM taxonomy on mycobiome diversity and composition; (ii) compare the mycobiome of individual gall compartments and investigate the involvement of neutral processes in fungal community assembly; and (iii) discuss the potential functional significance of the revealed fungal symbionts. We hypothesized that larvae would harbor fungal microbes different from nutritive mycelia, in analogy to detritophagous and herbivorous insects (33, 34, 43). Based on this assumption, presumed symbiosis with *B. dothidea,* and the active suppression of unsuitable taxa by larval symbionts (44), we expected a lower involvement of neutral processes in the gall interior and larval microbiome compared to the gall surface. Our sampling design enabled us to distinguish random fungal associates (plant endophytes and epiphytes) from real symbionts.

## RESULTS

From 31 triplets, 795 086 fungal reads were obtained. After discarding the contaminant and unassigned reads, 793 200 fungal reads ($\mu$ = 8529.03; 95% of them ranging from 1007 to 18 159) were assigned to 645 amplicon sequence variants (ASVs), which we classified into 184 species (gall surfaces = 112; gall interiors = 73; larvae = 127). On average, we observed 20.61 fungal species per sample ($\mu$ gall surfaces = 14.97 ± 5.12; $\mu$ interiors = 18.03 ± 5.19; $\mu$ larvae = 28.84 ± 7.05). For most samples, the sequencing depth was sufficient, as rarefaction curves at the ASV level reached their asymptotes (Fig. S1; Supplementary Material 1).

### Composition and diversity of mycobiomes

The gall compartment (i.e., gall surface, gall interior, and larva) best explained the mycobiome composition (13.86% of variability; df = 90, F = 10.03, $P$ = 0.001), followed by AGM species (16.28%; df = 85, F = 4.71, $P$ = 0.001) and their interaction (18.05%, df = 75, F = 2.61, $P$ = 0.001; Fig. 1), while the genus did not decrease Akaike information criterion (AICc). The results of this analysis at the ASV level were similar (Fig. S2; Supplementary Material 1). Analysis of each gall compartment separately revealed the greatest difference among individual species at the larvae level (AGM species explaining 47.68% of variability, df = 25, F = 4.56, $P_{adj}$ = 0.001), followed by the surface of the gall (that is, different host plants, 40.89% of variability, df = 25, F = 3.46, $P_{adj}$ = 0.001), while the lowest but still significant differences between AGM species were detected in the gall interior (30.73% of variability, df = 25, F = 2.22, $P_{adj}$ = 0.001). β-Diversity did not differ between the gall compartments (df = 88, F = 2.44, $P$ = 0.093).

The gall compartment was the only significant explanatory variable for the fungal richness (42.70% variability; df = 90, F = 35.55, $P$ < 0.001); neither the AGM genus (df = 89, F = 0.36, $P$ = 0.552) nor the species (df = 85, F = 0.82, $P$ = 0.516) further decreased AIC. The highest richness was associated with larvae (Fig. 2). Compared to the gall surface, the richness of the gall interior did not differ (df = 90, t = 0.50, $P$ = 0.620), while the larval richness differed significantly (df = 90, t = 6.93, $P$ < 0.001), as indicated also by the species accumulation curves (Fig. S3; Supplementary Material 1). The ASV-based analysis of richness yielded similar results (Fig. S4; Supplementary Material 1). The taxonomic diversity of the mycobiome differed significantly between the gall compartments (df = 90, F = 13.43, $P$ < 0.001), with gall interior having lower (t = −2.21, $P$ = 0.030) and larvae higher (t = 2.94, $P$ = 0.004) taxonomic diversity compared to surface mycobiome (Fig. 3).

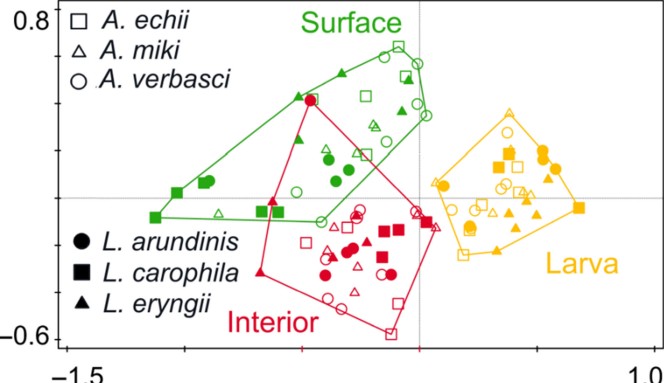

**FIG 1** Principal coordinates analysis plot based on partial-canonical correspondence analysis (F = 3.80, $P$ = 0.001) showing dissimilarity in the composition of fungal species among mycobiomes of gall surface, interior, and larvae of *Asphondylia* spp. and *Lasioptera* spp.

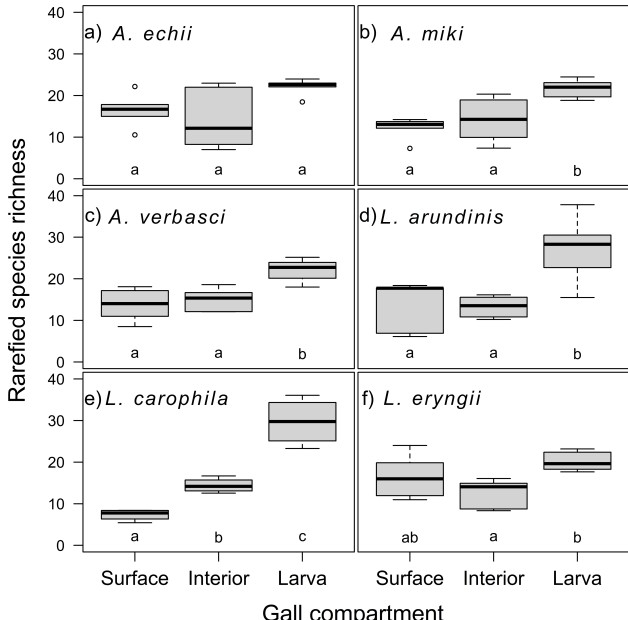

**FIG 2** Fungal species richness standardized by rarefaction for gall mycobiomes of individual gall parts for each gall midge species: (A) *Asphondylia echii*, (B) *A. miki*, (C) *A. verbasci*, (D) *Lasioptera arundinis*, (E) *L. carophila*, and (F) *L. eryngii*. Significant differences are indicated by different letters.

## Involvement of neutral processes in community assembly

The mycobiome was assembled most stochastically on the gall surface (89.00% of the ASVs fitted the prediction of the neutral model), while it was assembled least stochastically in the gall interior (77.80% of ASVs fitted the model prediction), and the result for the larvae was intermediate (84.40%, Fig. 4). We found 25 fungal taxa significantly indicative of larvae of at least one AGM species, six of which were common to three or more AGM species: *Fusarium sporotrichioides* (5 AGM species), *Filobasidium oeirense* (5), *Tilletiopsis washingtonensis* (3), *Alternaria* sp. (*A. alternata* species

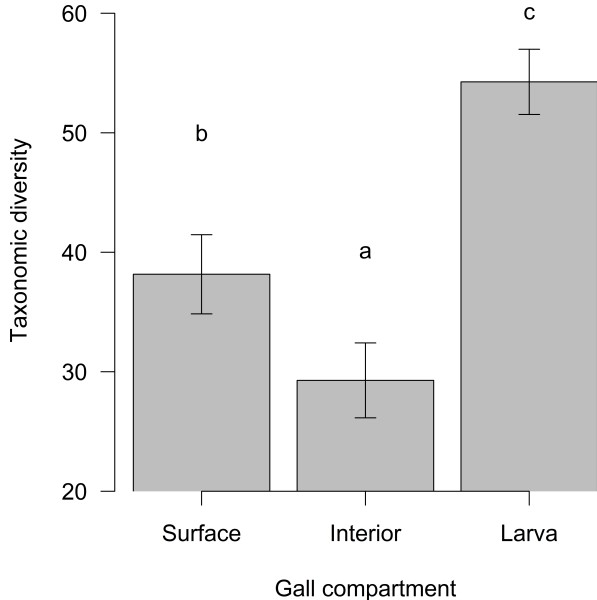

**FIG 3** Taxonomic diversity (Δ) of fungal species in individual gall compartments (mean ± SE). Significant differences are indicated by different letters.

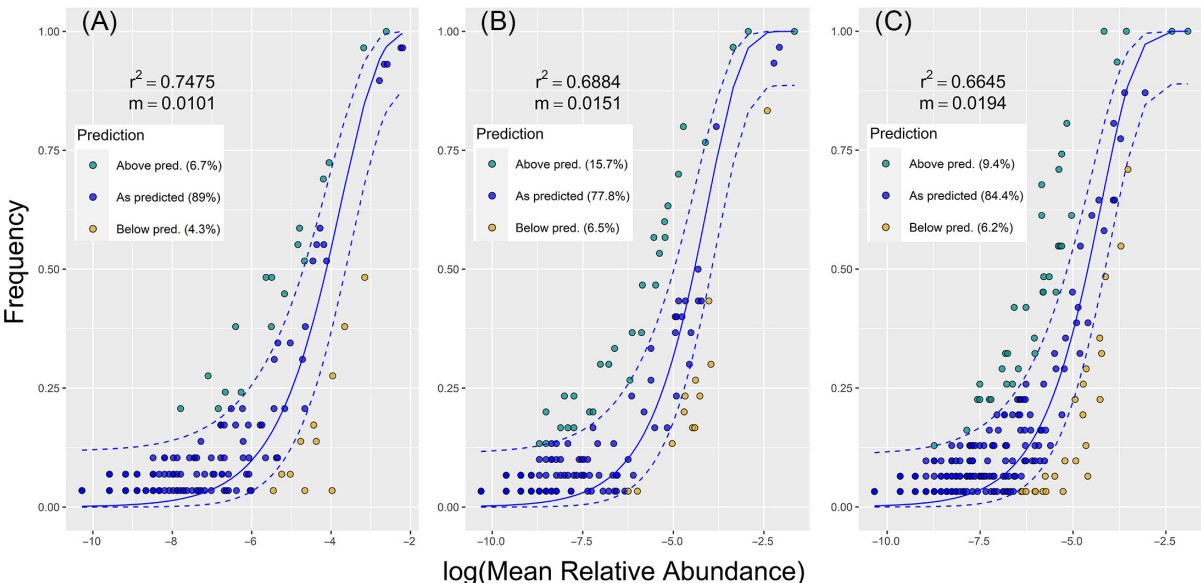

**FIG 4** Fit of the neutral community model (NCM) of fungal community assembly for (A) gall surface, (B) gall interior, and (C) larvae predicted occurrence frequencies for x and y. Solid blue lines indicate the best fit for the NCM according to Sloan et al. (45), and the dashed blue lines represent 95% confidence intervals (CIs) around the model prediction. Fungal species that occur more or less frequently than predicted by the NCM are shown in different colors. m = metacommunity size ×immigration; $r^2$ = the goodness of fit of this model (coefficient of determination).

complex) (3), unidentified *Alternaria* (3), and *Aureobasidium pullulans* (3). Only three fungal species were indicative of the gall interior: *Botryosphaeria dothidea* (4 AGM species), *Cercospora* sp. (*C. beticola* species complex) (1), and unidentified *Myrmecridium* species (1). *Botryosphaeria dothidea* was eudominant (>10%) in *A. echii* (80.80% of reads), *A. miki* (72.54%), *A. verbasci* (69.84%), and *L. eryngii* (74.84%), while in *L. carophila*

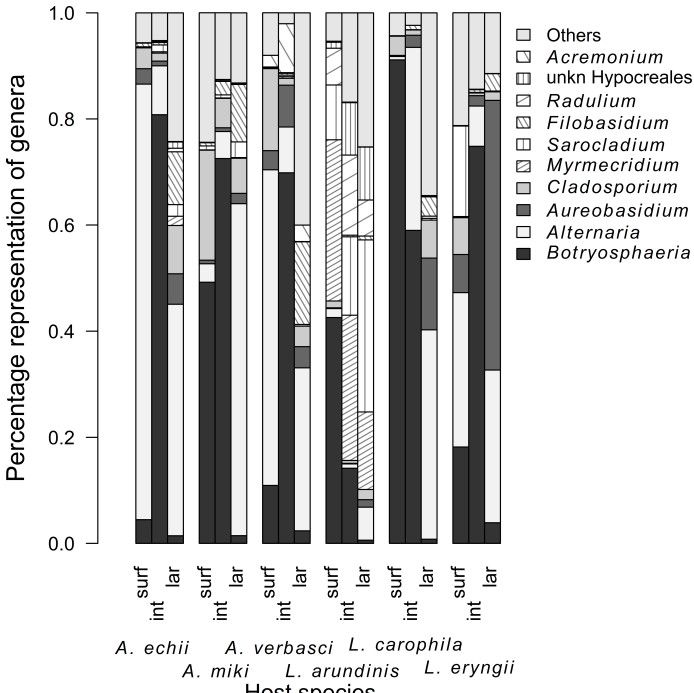

**FIG 5** Composition of mycobiomes at the genera level associated with the gall surfaces (S), gall interiors (I), and larvae (L) for individual *Asphondylia* and *Lasioptera* species.

(58.99%) it was accompanied by *Alternaria* sp. (*A. alternata* species complex) (31.26%). In *L. arundinis*, we found four eudominant species (>10%): unidentified *Myrmecridium* sp. (27.39%), *Radulidium subulatum* (15.11%), *B. dothidea* (14.16%), and *Sarocladium strictum* (11.01%). Five fungal species were indicative of gall surfaces and none of these were common among AGM species (Table 1; Fig. 5). The *ITS2* barcode did not provide sufficient discrimination power for some taxa, and the identities of significant taxa are provided in Supplementary Material 2B. Precise barcoding of the pure cultures confirmed the presence of *B. dothidea* and identified other dominant taxa, such as various species of *Akanthomyces, Aureobasidium, Alternaria, Cladosporium, Filobasidium, Fusarium, Heterophoma, Mucor, Papiliotrema, Rhodosporidium, Rhodosporidiobolus*, and *Sarocladium* (Supplementary Material 2C). As confirmed by the Venn diagrams, the pool

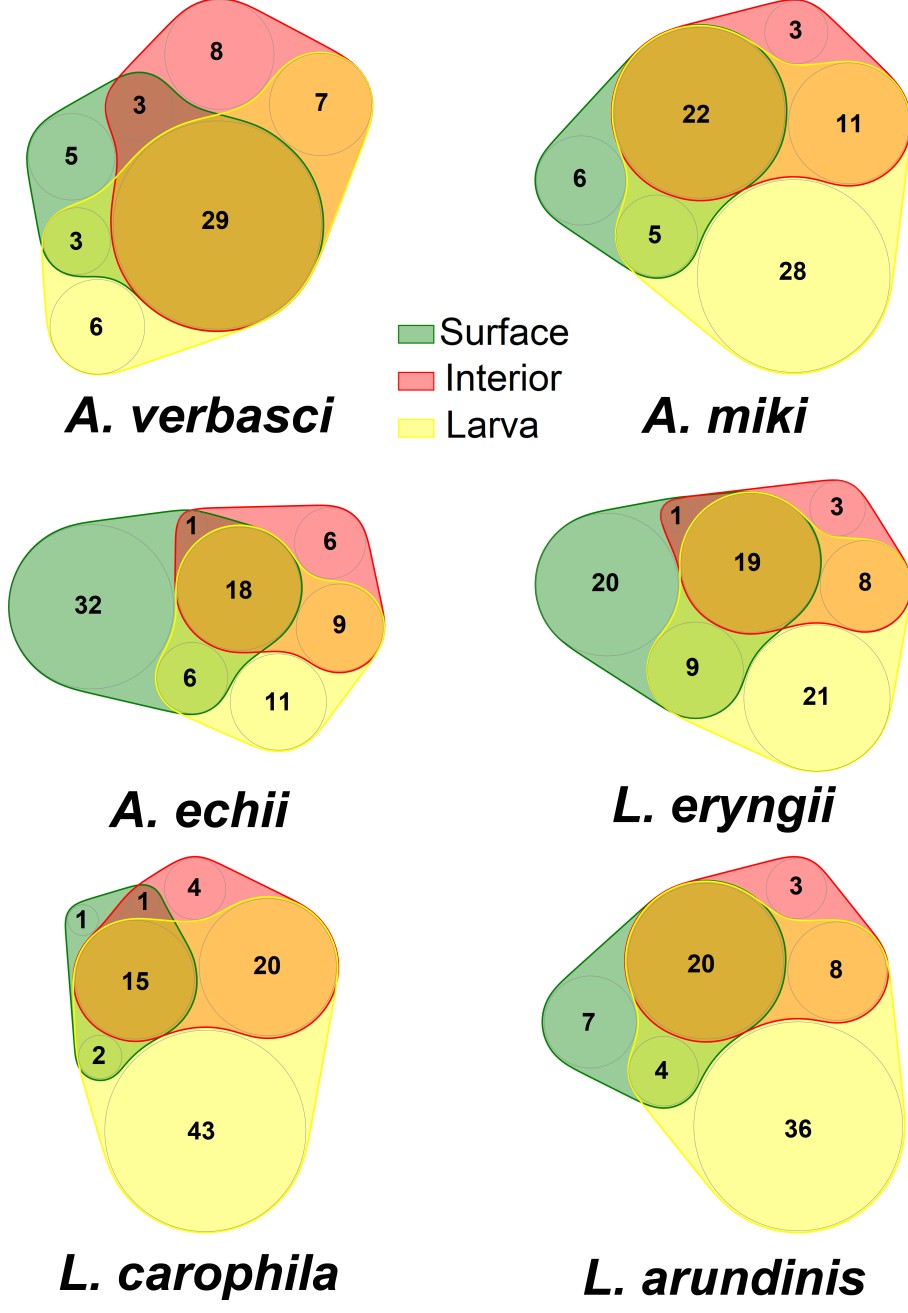

**FIG 6** Venn diagrams showing the overlap between mycobiome of gall surface (green), gall interior (red), and gall midge larvae (yellow) of individual *Asphondylia* and *Lasioptera* species.

**TABLE 1** Fungal species indicative of gall surfaces, gall interiors, and larvae for individual ambrosia gall midge (AGM) species mycobiomes based on MC statistics for multi-level pattern analysis ($P < 0.050$)[a]

| AGM species | Gall surfaces | Gall interiors | AGM larvae |
|---|---|---|---|
| *Asphondylia echii* | *Alternaria alternata* species complex | *Botryosphaeria dothidea* | *Filobasidium oeirense, Fusarium sporotrichioides, Fusarium avenaceum* species complex) |
| *Asphondylia miki* | | *B. dothidea* | *A. alternata* species complex, *Alternaria* sp., *Fil. oeirense, Fus. sporotrichioides, Neosetophoma* sp., *Periconia byssoides, Tilletiopsis washingtonensis* |
| *Asphondylia verbasci* | Botryosphaeriaceae sp., *Cladosporium* sp. 1, *Cladosporium* sp. 2 | *B. dothidea, Myrmecridium* sp. | *Fil. oeirense, Fil. wieringae, Aureobasidium pullulans, Ramularia* sp., *Rhodotorula glutinis* |
| *Lasioptera arundinis* | | *Cercospora. beticola* species complex | *A. alternata* species complex, *Alternaria* sp., *Apenidiella* sp. 1, *Apenidiella* sp. 2, *Aureobasidium pullulans, Fil. oeirense, Fus. sporotrichioides, A. pullulans, Mucor fragilis, Pseudopithomyces rosae, T. washingtonensis* |
| *Lasioptera carophila* | *B. dothidea* | | *Botrytis cinerea, Fus. sporotrichioides, Myrmecridium* sp., *Papiliotrema frias, T. washingtonensis* |
| *Lasioptera eryngii* | | *B. dothidea* | *A. alternata* species complex, *Alteranaria* sp., *Aur. pullulans, Cercospora zebrina, Cladosporium* sp. 2, *Fil. oeirense, Fil. wieringae, Fus. sporotrichioides, A. pullulans, Periconia* sp., *Pse. rosae,* Mycosphaerellaceae sp. |

[a]Full taxa names are listed in Supplementary Material 2B.

of fungi associated purely with larvae formed in a total of 27.87% of taxa and was the largest subset in four of six gall midge species (Fig. 6).

## DISCUSSION

Larval mycobiomes differed significantly in composition from the surface, and gall interior mycobiomes were the most species-rich and taxonomically diverse. They hosted the largest pool of significantly associated and species-specific taxa. These results indicate that the larval interior is a prolific environment that supports the growth of diverse and host-specific fungal consortia. In accordance with our hypothesis, the mycobiome of the gall surfaces was assembled most stochastically, while the involvement of neutral processes was the lowest in the mycobiomes of the gall interiors, which also had the lowest taxonomic diversity. The fungal composition in nutritive mycelia tended to be more similar across AGM species than larval and gall surface mycobiomes, suggesting that the interior of ambrosia galls offers a unique environment that supports the growth of similar fungal groups, despite the phylogenetic distances between the host plant species and between the AGM tribes Asphondyliini and Lasiopterini.

The higher similarity of the mycobiomes of the gall interiors between species was due to the dominance of *Botryosphaeria* (except in *L. arundinis*). In our study, the sequenced ITS2 regions assigned to *Botryosphaeria* were identical to the type strain of *B. dothidea* and conspecific isolates from gall midges (12, 13, 20, 27, 46). This marker is not specific to distinguish it from related species such as *B. auasmontanum, B. fabicerciana, B. scharifii, B. ramosa,* and *B. fusispora.* All these species were not reported as AGM symbionts and were known only from outside Europe (47) (Table S1; Supplementary Material 2A). Finally, our taxonomic analysis using ITS rDNA and TEF1α barcode on pure cultures confirmed its identity as *B. dothidea.*

The dominance of *B. dothidea* as an AGM symbiont aligns with Bisset and Borkent (2), who suggested that there might be fewer species of fungi than cecidomyiids involved in ambrosia galls, with several midges utilizing the same dominant fungus (12). Asphondyliini may depend on one principal fungus, primarily *B. dothidea,* sharing specific clonal strains among several species, supplied by some nonspecific fungi (2, 12–15, 18, 48, 49). *Botryosphaeria dothidea* can also be found in other AGMs, including Lasiopterini (50) and

Alycaulini (12), but the specificity in Lasiopterini has received much less attention. As Asphondyliini and Lasiopterini form distant clades of Cecidomyiidae, a long evolutionary association with gall midges is likely for *B. dothidea* (2, 26).

In *L. arundinis* and *L. carophila*, *B. dothidea* was more prevalent on the gall surfaces than in the gall interiors (42.56% and 91.13% of reads, respectively). *Botryosphaeria dothidea* is a ubiquitous endophyte and pathogen that occurs in multiple plants worldwide (51). Facilitated colonization of endophytes in galled plant tissue is known for non-AGMs (52, 53). It is possible that in AGMs, endophytes may have become fungal symbionts, being genetically identical to free-living populations (17). However, the midge *Asteromyia carbonifera* (Osten Sacken 1862; tribe Alycaulini, supertribe Lasiopteridi) appears to be associated with a single lineage of *B. dothidea* despite the supposed lack of vertical transmission. This symbiosis exhibits an unusually high level of specificity for ectosymbiotic associations (12), which are prone to invasion and replacement of symbionts (17, 54). Therefore, *B. dothidea* may be vertically transmitted by mycangia (17) or acquired environmentally by horizontal transfer in each generation, as is known in fungal symbionts of termites and some wood wasps (55, 56). In such cases, microbial competition among potential symbionts may sustain the specificity of symbiosis through competition-based selection, potentially because hosts provide a specific environment to selectively cultivate favorable symbionts (57). AGM larvae have been shown to inhibit the growth of competing fungi (15, 58) or regulate the metabolism of *B. dothidea* because the virulence of pathogenic *B. dothidea* is suppressed in the galls (18, 48). *Botryosphaeria dothidea* itself may also produce metabolites with antimicrobial or antifungal activities (59); therefore, *B. dothidea* and larvae may protect each other from unwanted invaders of the gall. Competition-based selection or active larval selection can be corroborated by a reduced role of neutral processes in the gall interior compared to other gall compartments.

In the interior of *L. arundinis* galls, the dominant fungal species were *Myrmecridium* sp., *R. subulatum*, and *Sarocladium strictum*. Furthermore, a significantly indicative taxon was assigned to the genus *Cercospora*. The association between *L. arundinis* and *R. subulatum*, previously classified as *Ramichloridium* sp., is already recognized (2, 60). A closely related gall midge, *L. hungarica,* exhibits a similar relationship with a fungus named *Sporotrix* sp., probably also conspecific with *R. subulatum* (61). *Cercospora* is a plant pathogen, endophyte, and saprobe. The same applies to *Sarocladium* (62–64)*, which was also found in the galls of *Lasioptera donacis* (22). Species from the genus *Myrmecridium* can be found as saprobes (64). Other dominant fungi in the gall interior were *Alternaria* and *Aureobasidium*. *Alternaria* has been reported in mycangia or galls of various *Asphondylia* species, sometimes also as a prominent fungus (12, 20, 28, 65). *Aureobasidium* covers the interior of the galls of some *Asphondylia* (24, 66) or *Lasioptera* (67). Both *Alternaria* and *Aureobasidium* are known as plant endophytes (68, 69) and may also have a nutritive role (21, 70), and both were predominant in larval mycobiomes.

The larval mycobiome was likely composed mainly of the taxa present in the gut. It is not clear whether the fungi occurred in the internal organs or the hemolymph, but the effect of surface contamination was likely very low given the distinct composition of the mycobiome of the larvae and the gall interior and the fact that the larvae were surface-sterilized. In Cecidomyiidae, data on the bacterial part of the intestinome are only available (36, 39), although in fungus-feeding insects, gut microbiomes may help digest fungal biomass (37, 71). In our study, the AGM gut mycobiomes differed substantially from the gall interior; moreover, there was a significant effect of AGM species on the fungal community composition. The significant effect of the host species on the mycobiome composition is contradictory to the patterns found in larval leaf miners (also concealed within the plant tissue but not fungal feeders (43), but similar to those found in leaf-chewing Lepidoptera (33). As the extent to which AGM larvae feed on plants, except nutritive mycelia, remains almost unknown (72, 73), it is possible that these associates are acquired from the host plant and subsequently filtered by the gut environment, which is usually species-specific (33).

Based on previous studies, the core mycobiome of AGM galls is largely composed of saprotrophs (27) and sometimes pathogenic fungi widespread in plants (12, 46), growing epiphytically or endophytically (27, 74). These taxa are good biomass degraders and are tolerant to extreme conditions and stress, they grow rapidly and sporulate. This may also apply to certain taxa indicative of larvae in our study, such as several *Fusarium* species. In addition to being pathogenic to plants (75), *Fusarium* can play a role in the defense of insects against pathogens (76), as many strains produce a wide variety of mycotoxins (77), many of which have antibacterial activity (78, 79). Insect vectoring of plant pathogens after ingestion can evolve into mutualism if the insect benefits from the plant infection (80). On the other hand, *Fusarium* spp. are often mentioned as entomopathogens whose mycotoxins can have a noxious effect on the larvae rather than protecting them from infections (81).

A similar spectrum of genera found in the larvae in this study (*Alternaria*, *Fusarium*, *Cladosporium*, and various yeasts, including *Aureobasidium*) has been found in the guts of other plant biomass feeders. Among the taxa recorded in this study as significantly indicative of larvae, *T. washingtonensis* was found in the gut mycobiomes of herbivorous beetles (82), *Fil. oeirense* and *Pseudopithomyces* occurred in the gut mycobiomes of Tephritidae fruit flies (83, 84) and *Filobasidium* and *Didymella* in the larval gut myco-biomes of ambrosia beetles (41, 42). Interestingly, *T. washingtonensis* and *A. pullulans* are probably capable of inducing gall formation (67, 85), and Didymellaceae and *Filobasi-dium* have been reported from various galls and gall formers (85–87). *Neosetophoma* is an anamorph of *Didymella*, which is known to infect the feeding sites of some Cecidomyiidae (88), and *Fusarium* has been recorded in the galls of some *Daphnephila* (Cecidomyiidae: Asphondyliini) (14, 48) and *Asphondylia* (20) species. The secretions produced by AGM larvae are believed to be responsible for gall development (2, 5), and the aforementioned fungi may be involved in the formation of these secretions. Galling insects have been proposed to mediate gall induction by endosymbiotic microorganisms or gall-inducing genes acquired from microbial symbionts through horizontal gene transfer (89, 90).

The larvae likely affect the growth of the fungi in nutritive mycelia because the main fungal development and changes in color and rigidity are observed when larvae stop feeding, die, or are attacked by a parasitoid (13, 15, 16, 23, 91). It is possible that the enzymes secreted by larvae can help modify the composition of the nutritive mycelia. We hypothesized that some of the fungi indicative of larvae might play a role in producing such enzymes. Antibiotic-producing microbes that defend fungal gardens from antagonistic organisms occur in other fungus-growing insects (92–97), and the convergent evolution paradigm may suggest the presence of particular antibiotic-pro-ducing microbes in gall midges (98). Microbes with symbiotic functions were found in the bacteriocyte-like structure of eggs transmitted maternally in distantly related midges (99). Thus, larval fungal symbionts may be included in further research on potential endosymbiotic microorganisms in AGM species.

## Conclusion

For the first time in gall-forming insects, we present a complete characterization of the mycobiomes of the whole system as we profiled gall surfaces, nutritive mycelia, and larvae. Our study suggests a spectrum of fungal microorganisms indicative of individual gall compartments. We have discovered that the most diverse and unique communities are associated with hitherto unstudied intestinal mycobiomes of larvae. However, in AGM larvae, the specificity and role of these fungi remain unresolved. As antibiotic and antifungal properties may be found in these endosymbionts during further research, congruent with the convergent evolution paradigm, AGM larvae could have considerable biochemical potential.

## MATERIALS AND METHODS

### Sampling and processing of larvae and plant tissues

Galls were sampled in the Cerová vrchovina Protected Landscape Area (48.219N, 19.967E, Slovakia) in August 2019. We sampled six AGM species: Asphondyliini: *Asphondylia echii* (Loew 1850) from *Echium vulgare* L., *A. miki* Wachtl, 1880 from *Medicago falcata* L., *A. verbasci* (Vallot 1827) from *Verbascum* sp. L.; Lasiopterini: *Lasioptera arundinis* Schiner (1854) from *Phragmites australis* (Cav.) Trin. ex Steud., *L. carophila* Löw (1874) from *Daucus carota* L., and *L. eryngii* (Vallot 1829) from *Eryngium campestre* L. The galls were placed in separate plastic containers using sterilized tweezers to avoid contamination. To obtain the microbiota of the gall surface (plant tissue), the surface tissue of the galls was scratched with a sterilized razor blade (approx. thickness 0.1 mm). The galls were then washed by vortexing in a 30-mL centrifuge tube with a 20-mL sterile solution of phosphate-buffered saline (PBS) 1% with Tween 80 (Sigma-Aldrich, Saint Louis, MI, USA) at 2,100 rpm for 45 s to minimize contamination during dissection (100). The galls were dissected on paraffin wax, which was previously sterilized by pouring ethanol and igniting. For subsequent experiments, we used galls with one living larva inside for *Asphondylia* and at least one living larva inside for *Lasioptera* (empty galls and galls with pupae, dead larvae, or larvae with parasitoids were discarded). A total of 31 galls were used (5–6 galls from each species): *A. echii* = 5, *A. miki* = 6, *A. verbasci* = 6, *L. arundinis* = 5, *L. carophila* = 4, *L. eryngii* = 5. We scratched their interior and separated one larva. The larvae were washed in the same manner as the galls to minimize contamination by the gall interior. The gall surface, gall interior, and larva represented triplets in subsequent analyses ($n = 31 \times 3$ gall compartments).

### Metabarcoding of mycobiomes

Total microbial DNA was extracted from the samples using the NucleoSpin Tissue DNA Isolation Kit, following the manufacturer's protocol with minor modifications. For a complete lysis of the samples and higher DNA yields, we crushed the samples multiple times in 1.5 mL tubes using pestles and liquid nitrogen before the cell lysis step. To ensure the broad recovery of fungal diversity and to significantly reduce the recovery of chloroplasts, we used highly degenerate primers to amplify the *ITS2* rDNA variable gene regions. All PCRs were performed in triplicate. We used the primers ITS3_KYO2 5′–GATGAAGAACGYAGYRAA–3′ (forward) and ITS4_KYO3 5′–CTBTTVCCKCTTCACTCG–3′ (reverse) (101), with barcodes added to the 5′ end of both primers, enabling us to identify each sample. Amplification was performed as described by Toju et al. (101), with minor modifications consisting of initial denaturation at 95℃ for 3 min; 35 cycles at 94℃ for 30 s, 55℃ for 60 s, and 72℃ for 60 s; and a final extension at 72℃ for 10 min. Each PCR (25 µL) consisted of 9.4 µL molecular biology grade water (New England BioLabs), 0.5 U KAPA2G Robust HotStart DNA Polymerase, 5 µL 5 × KAPA2G Buffer B, 5 µL 5 × KAPA2G Enhancer (all Kapa Biosystems), 0.5 µL 10 mM dNTP Mix (Thermo Fisher Scientific), 0.8 µM of each primer, and 2 µL genomic DNA. On the plate ($n = 96$), negative controls ($n = 3$) (mastermix +water + primers) were placed evenly. All PCR products were checked on a 1.5% agarose gel. Subsequently, we pooled triplicate PCRs within each sample, measured DNA concentration using the Qubit dsDNA BR Assay Kit (Thermo Fisher Scientific), and equalized concentrations within all samples. Furthermore, the samples were pooled to create a library. Amplicons of specific length were excised from the 2% agarose gel and purified using QIAquick Gel Extraction Kit (Qiagen) and subjected to DNA ligation of sequencing adapters and library-unique multiplex identifiers necessary for demultiplexing the reads using the KAPA Hyper Prep Kit (Kapa Biosystems) following the manufacturer's instructions. The ligated library was quantified using the KAPA Library Quantification Kit (Kapa Biosystems) and diluted to create a final sequencing library at 7.5 ng/µL. The library was subjected to paired-end sequencing on an Illumina MiSeq instrument at the Genomics Core Facility, CEITEC (Masaryk University, Brno, Czech Republic), producing 2

× 300 bp long reads (four runs in total). Sequences dedicated to this study represented 4.68% of the total sequencing output.

## Metabarcoding data processing

Sequencing data were processed using QIIME 2.0 2020.2 (102). Raw paired-end reads were demultiplexed and quality filtered, including extraction of the *ITS* region using the q2-ITSxpress plugin (103). Reads were denoised using the DADA2 algorithm (104). Taxonomy was assigned using the q2-feature-classifier classify-sklearn (105) using a trained naïve Bayes classifier against the reference sequences in the UNITE QIIME release for Fungi version 8.0 (106). Information on the read counts for all amplicon sequence variants (ASVs) from all samples together with taxonomic information was compiled into the ASV table.

## Statistical analysis

All analyses were performed in R 4.2.1 (107) and Canoco 5.01 (108). We identified the contaminant ASVs based on negative controls using the library "decontam" (109) and discarded them. We then pooled the identified ASVs at the level of fungal species to perform the analyses, but the major analyses were repeated also at the level of ASVs to validate the patterns with a more detailed resolution (Figs S1, S2, and S4; Supplementary Material 1). We analyzed the differences in composition by Permutation Multivariate Analysis of Variance (PERMANOVA) with Bray-Curtis distance matrices and tested 999 permutations, using the library "vegan" (110). As potential explanatory variables, we used the triplet ID (if this was selected based on the AICc, it would be used as a random term in the final model), gall compartment, and taxonomy of AGM (species or genus). We built the final model by forward selection based on the AICc. We accompanied this analysis with canonical correspondence analysis (CCA) using the gall compartment as an explanatory variable and gall species as a covariate and testing the significance using the Monte Carlo test with 999 permutations. As the results of PERMANOVA can be compromised in case of an unbalanced number of samples, we added the PERMDISP2 procedure for the analysis of multivariate homogeneity of group dispersions (variances) based on the Bray–Curtis distance, measuring the distance to the group centroids (111). Differences in β-diversity among gall compartments were tested using ANOVA and Tukey HSD post hoc test. Based on our results, we repeated the PERMANOVA analysis for each gall compartment separately, with AGM species as an explanatory variable.

To analyze which explanatory variables best explained the richness, we used a generalized linear model (GLM) with Gamma distribution with forward selection based on AICc. Before analysis, we standardized the species richness of all samples by rarefying/extrapolating the read counts to a uniform value ($n = 1,000$) using the library "iNEXT" (112, 113). To analyze taxonomic diversity, we calculated an index developed by Clarke and Warwick (114, 115) (Δ; the average taxonomic distance between any two organisms, randomly chosen from the sample). Six levels of taxonomic resolution were used for index calculations: phylum, class, order, family, genus, and species. For analysis of taxonomic distance, we built GLM with Gamma distribution with forward selection based on AICc from previously mentioned explanatory variables. We accompanied the analyses with species accumulation curves (116) to assess the sufficiency of the sampling effort and with Venn diagrams created using a simulation-based nVenn algorithm from the library "nVennR" (117) for a simple display of the similarity of the gall compartments.

To quantify the involvement of neutral processes in the fungal community assembly, we fitted neutral models at the ASV level for each gall compartment according to Sloan et al. (45) using libraries "reltools" (118), "phyloseq" (119), and "GUniFrac" (120). First, we rarefied samples to the same sequence depth, i.e., 1,000 reads. Then, we fitted the neutral models and extracted information about taxa fitting the null model, being above or below prediction. At the level of fungal species, we identified the indicator species for each gall compartment by IndVal, the indicator value relating to the frequency and

relative abundance of the reads (121), and multilevel pattern analysis from libraries "indicspecies" and "labdsv" (122, 123).

## Cultivation and identification of fungi

Because in some fungal taxa, ITS2 generated by metabarcoding does not allow accurate taxonomic determination, we directly cultivated fungi from the galls and larvae of *A. echii*, *A. miki*, *A. pruniperda*, *A. verbasci*, *Lasioptera artemisiae*, *L. arundinis*, *L. carophila*, *L. eryngii,* and *L. rubi*. Small fragments of ambrosial fungal mat (1 × 1 mm) and individual larvae were resuspended in 1 mL of 1% PBS solution and crushed with a sterile stamen. One hundred thirty-two micorliters of the suspension (undiluted and 10-fold diluted) was subsequently spread evenly on agar plates with Malt Extract Agar (HiMedia, Mumbai, India). Agar plates were cultivated at 25°C for one week in the dark. After this period, colonies were morphotyped, and morphologically unique cultures were identified. Fungi were identified by ITS rDNA region (primers ITS1 and ITS4) and TEF1α region (primers EF 526F and 986R) barcode according to Kolařík et al. (124). Sequences were compared with data deposited in the NCBI database using the BLASTn tool, with a preference for data obtained from type cultures or reliable taxonomic studies.

## ACKNOWLEDGMENTS

We thank all colleagues from the Laboratory of Insect Trophic Strategies at the University of Ostrava for laboratory processing. We thank Soňa Kajzrová and Kateřina Křížková for assistance with fungal cultivation and identification.

This work was supported by the Czech Science Foundation (GA23-07026S).

P.D., P.P., and M.Š. designed the research; P.P. and J.Š. sampled the data; P.P., D.V., and M.Š. performed the laboratory work and databased the data; M.Kos. and M.Kol. processed the data bioinformatically; P.P. performed the statistical analysis; P.P. and H.Š. wrote the manuscript; and M.Š., M.Kol., J.Š., and P.D. edited the manuscript and provided additional comments.

## AUTHOR AFFILIATIONS

[1]Department of Biology and Ecology, Faculty of Science, University of Ostrava, Ostrava, Czech Republic
[2]Department of Zoology, Faculty of Science, Palacký University, Olomouc, Czech Republic
[3]Institute of Microbiology, Academy of Sciences of the Czech Republic, Prague, Czech Republic

## AUTHOR ORCIDs

Hana Šigutová  http://orcid.org/0000-0003-1134-248X
Miroslav Kolařík  http://orcid.org/0000-0003-4016-0335

## FUNDING

| Funder | Grant(s) | Author(s) |
| --- | --- | --- |
| Grantová Agentura České Republiky (GAČR) | GA23-07026S | Petr Pyszko |
| | | Hana Šigutová |
| | | Miroslav Kolařík |
| | | Martin Kostovčík |
| | | Jan Ševčík |
| | | Martin Šigut |
| | | Denisa Višňovská |
| | | Pavel Drozd |

## AUTHOR CONTRIBUTIONS

Petr Pyszko, Conceptualization, Data curation, Formal analysis, Investigation, Methodology, Visualization, Writing – original draft, Writing – review and editing | Hana Šigutová, Writing – original draft, Writing – review and editing | Miroslav Kolařík, Data curation, Project administration, Writing – review and editing | Martin Kostovčík, Data curation, Writing – review and editing | Jan Ševčík, Investigation, Writing – review and editing | Martin Šigut, Conceptualization, Data curation, Investigation, Methodology, Writing – review and editing | Denisa Višňovská, Investigation, Writing – review and editing | Pavel Drozd, Conceptualization, Methodology, Project administration, Writing – review and editing

## DATA AVAILABILITY

Raw demultiplexed sequencing data with sample annotations are available under the accession number PRJNA693163 at the NCBI Bioproject website.

## ADDITIONAL FILES

The following material is available online.

### Supplemental Material

**Supplemental Material S2A (Spectrum02830-23-s0001.docx).** Taxonomic identity of ITS2 sequences ascribable to the *Botryosphaeria* genus.
**Supplemental Material S2C (Spectrum02830-23-s0002.docx).** Taxonomic identity of cultures isolated from gall interior and larvae.
**Supplemental Material S1 (Spectrum02830-23-s0003.docx).** Figures S1 to S4.
**Supplemental Material S2B (Spectrum02830-23-s0004.docx).** Taxonomic identity of species significantly indicative for gall surfaces, gall interiors, and AGM larvae.

### Open Peer Review

**PEER REVIEW HISTORY (review-history.pdf).** An accounting of the reviewer comments and feedback.

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
