## [Reviewer comments · Microbiology Spectrum]

Microbiology Spectrum

Mycobiomes of two distinct clades of ambrosia gall midges (Diptera: Cecidomyiidae) are species-specific in larvae but similar in nutritive mycelia

Petr Pyszko, Hana Šigutová, Miroslav Kolařík, Martin Kostovčík, Jan Ševčík, Martin Šigut, Denisa Hařovská, and Pavel Drozd

Corresponding Author(s): Petr Pyszko, Ostravska univerzita

Review Timeline:

Submission Date:	July 11, 2023
Editorial Decision:	October 16, 2023
Revision Received:	October 23, 2023
Accepted:	October 24, 2023

Editor: Christina Cuomo

Reviewer(s): Disclosure of reviewer identity is with reference to reviewer comments included in decision letter(s). The following individuals involved in review of your submission have agreed to reveal their identity: Shu Benshui (Reviewer #1); Rosario Nicoletti (Reviewer #2)

Transaction Report:

DOI: <https://doi.org/10.1128/spectrum.02830-23>

October 16, 2023

Dr. Petr Pyszko
Ostravska univerzita
Ostrava
Czech Republic

Re: Spectrum02830-23 (Mycobiomes of two distinct clades of ambrosia gall midges (Diptera: Cecidomyiidae) are species-specific in larvae but similar in nutritive mycelia)

Dear Dr. Petr Pyszko:

Thank you for submitting your manuscript to Microbiology Spectrum. Two reviewers have provided feedback that I would like you to address in a revision. Please also ensure you have added a Data accessibility paragraph (<https://journals.asm.org/open-data-policy>) that includes accessions for your sequence data.

Link Not Available

Sincerely,

Christina Cuomo

Journals Department
Reviewer comments:

Reviewer #1 (Comments for the Author):

The authors investigated a complete characterization of the mycobiomes of the whole system as we profiled gall surfaces, nutritive mycelia, and larvae. The methods used in this study were relatively reasonable, and the results displayed supported the conclusions. However, some comments existed and showed as follows:

1. English is modest. Therefore, the authors need to improve their writing style. In addition, the whole manuscript needs to be checked by native English speakers.

2. The gall surfaces, gall interiors, and larvae should be displayed as a figure.
3. The raw data obtained by sequencing should be submitted to the public repository.
4. Why the authors used ITS3 and ITS4 primers to amplify the ITS2 rDNA?
5. The Figure 5 should be revised, the horizontal coordinate is chaotic.

Reviewer #2 (Comments for the Author):

This manuscript reports original data on fungi associated with a few species of cecidomyid gall midges. Although the authors could not demonstrate a clear symbiotic role by any of the identified species, their findings represent a valuable contribution for improving our understanding of this unique biological association. The manuscript is well written and offers an exhaustive introductory overview of the current knowledge on the subject. I only observe that the discussion section should be revised by removing some uncircumstantial considerations. Particularly, concerning text at lines 394-398 I remark that *Fusarium* spp. are very diverse in their occurrence and ecological interactions; in fact, they are often mentioned as entomopathogens, and their mycotoxins could have a noxious effect on the larvae rather than protecting them from bacterial infections. Some statements in this section also require revision. At line 366, it is incorrect that *Radulidium subulatum* had been previously identified as *Macrophoma* sp.; indeed, *Macrophoma* is an old name for *Botryosphaeria dothidea*, and it is very likely that fungi identified with this name in the old papers corresponded to the latter species. At line 368-369 it is incorrect that *Cercospora* (Dothideomycetes) is similar to *Sarocladium* (Sordariomycetes). Finally, at lines 370, 'plant' should be deleted; in fact, the saprobic aptitude is not intended to be exerted on living organisms, and ref. 89 generically qualifies *Myrmecridium* as 'saprobes'.

Staff Comments:

Preparing Revision Guidelines

Please return the manuscript within 60 days; if you cannot complete the modification within this time period, please contact me. If you do not wish to modify the manuscript and prefer to submit it to another journal, please notify me of your decision immediately so that the manuscript may be formally withdrawn from consideration by Microbiology Spectrum.

23 October 2023
Dr Christina Cuomo
Editor, Microbiology Spectrum

Dear Dr Cuomo,

Thank you for considering our manuscript entitled “**Mycobiomes of two distinct clades of ambrosia gall midges (Diptera: Cecidomyiidae) are species-specific in larvae but similar in nutritive mycelia**” (02830-23) for publication. I, along with my co-authors, would like to re-submit its revised version.

We received two very positive reviews on our manuscript. We carefully checked the manuscript and made appropriate changes in accordance with the reviewers’ suggestions. The manuscript has also been checked by the native English speaker (in accordance with the reviewer 1 comment). We believe that the comments of both reviewers have improved the quality of our manuscript. Our responses are attached herewith.

We look forward to hearing from you regarding our submission. We would be glad to respond to any further questions and comments that you may have.

Sincerely,

Hana Šigutová

Reviewer #1 (Comments for the Author):

English is modest. Therefore, the authors need to improve their writing style. In addition, the whole manuscript needs to be checked by native English speakers.

We have taken your comment into consideration and made several improvements to the language. Specifically, we have addressed typographical errors, included missing articles, improved sentence punctuation, and replaced inappropriate prepositions. We believe that these changes have enhanced the overall quality of the text. Moreover, the manuscript has been re-checked by native English speaker.

The gall surfaces, gall interiors, and larvae should be displayed as a figure.

We are not sure we understand this comment, but the differences in microbial communities among individual gall parts are displayed in Figures 1–5.

The raw data obtained by sequencing should be submitted to the public repository.

Raw data are deposited in a public repository, the information is provided in the manuscript (now newly as a Data availability statement, lines 411–414). We also added the direct website link.

Why the authors used ITS3 and ITS4 primers to amplify the ITS2 rDNA?

These are standard primers used to amplify ITS2 rDNA region. Based on our previous experience, by combination of these primers it is possible to obtain the highest possible coverage (e.g., <https://www.nature.com/articles/s41598-022-19855-5>; <https://journals.asm.org/doi/full/10.1128/spectrum.03160-22>).

The Figure 5 should be revised, the horizontal coordinate is chaotic.

Thank you for this comment. We altered the colors and patterns in the figure, and we added a better caption of axes. We believe that the whole figure looks less chaotic now.

Reviewer #2 (Comments for the Author):

This manuscript reports original data on fungi associated with a few species of cecidomyid gall midges. Although the authors could not demonstrate a clear symbiotic role by any of the identified species, their findings represent a valuable contribution for improving our understanding of this unique biological association. The manuscript is well written and offers an exhaustive introductory overview of the current knowledge on the subject. I only observe that the discussion section should be revised by removing some uncircumstantial considerations. Particularly, concerning text at lines 394-398 I remark that *Fusarium* spp. are very diverse in their occurrence and ecological interactions; in fact, they are often mentioned as entomopathogens, and their mycotoxins could have a noxious effect on the larvae rather than protecting them from bacterial infections.

Thank you for this comment; we incorporated this important information to the manuscript (lines 253–255).

Some statements in this section also require revision. At line 366, it is incorrect that *Radulidium subulatum* had been previously identified as *Macrophoma* sp.; indeed, *Macrophoma* is an old name for *Botryosphaeria dothidea*, and it is very likely that fungi identified with this name in the old papers corresponded to the latter species.

Thank you for this comment, corrected (line 220).

At line 368-369 it is incorrect that *Cercospora* (Dothideomycetes) is similar to *Sarocladium* (Sordariomycetes).

Thank you for this comment; we agree. This statement was misinterpreted due to incorrect translation – we corrected it (line 223).

Finally, at lines 370, 'plant' should be deleted; in fact, the saprobic aptitude is not intended to be exerted on living organisms, and ref. 89 generically qualifies *Myrmecridium* as 'saprobes'.

Deleted (now line 225).

Re: Spectrum02830-23R1 (Mycobiomes of two distinct clades of ambrosia gall midges (Diptera: Cecidomyiidae) are species-specific in larvae but similar in nutritive mycelia)

Dear Dr. Petr Pyszko:

Your manuscript has been accepted, and I am forwarding it to the ASM production staff for publication. Your paper will first be checked to make sure all elements meet the technical requirements. ASM staff will contact you if anything needs to be revised before copyediting and production can begin. Otherwise, you will be notified when your proofs are ready to be viewed.

Sincerely,
Christina Cuomo
Editor
Microbiology Spectrum